# Imbibition, Germination, and Early Seedling Growth Responses of Light Purple and Yellow Seeds of Red Clover to Distilled Water, Sodium Chloride, and Nutrient Solution

**Ana Sofia Costa, Luís Silva Dias \*** and **Alexandra Soveral Dias**

Department of Biology, University of Évora, Ap. 94, 7002-554 Évora, Portugal
* Correspondence: lsdias@uevora.pt

**Abstract:** The seeds of red clover are heteromorphic and two color morphs can be visually recognized, light purple and yellow, resulting from heterozygosity and recessive homozygosity at two loci. Here, we report the responses of seed imbibition, seed germination, and early seedling growth of the two morphs to distilled water, sodium chloride, and complete nutrient solution. The sensitivity of red clover seeds to treatments increased with the stage of development in what seems to be a cumulative process. No differences were found in seed imbibition between morphs or between treatments. In seedling growth, on the contrary, treatments were always effective, but differences between morphs were only observed in seeds that were treated with nutrient solution, whereas in the intermediate stage of seed germination, the effects by treatments were observed together with the appearance of differences between morphs in distilled water and in the treatment by sodium chloride solution. Simultaneously, the superior performance of the yellow morph that was found in germination, which appears to be a trait stable across cultivars of red clover seeds, turned into a superior performance of the light purple morph in seedling growth.

**Keywords:** germination; heteromorphism; imbibition; seed color; seedling growth; sodium chloride; *Trifolium pratense*

---

## 1. Introduction

Despite having lost much of the agricultural importance that it once had, mostly because of the widespread usage of synthetic fertilizers from the 1950s onwards [1], red clover (*Trifolium pratense*) is still widely used worldwide, especially in temperate zones [1,2].

Red clover has been the target of continuous efforts of breeding, especially in longevity, which primarily resulted in the increase from two to four years of its persistence in crop systems. Indirectly, breeding for increased longevity has also resulted in better seedling establishment through increased densities of plant establishment [1,3].

The seeds of red clover are produced in a variety of colors, some of which are genetically controlled and others, brown ones, result from the ageing of seeds or from unfavorable conditions during production or storage [4–7]. In red clover seeds, colors are located in the seed coat not in the endosperm or embryo, and therefore the mother plant and not the seed is the unit for the inheritance of color [6,8]. Seed coat color is controlled in two loci, with yellow morphs being homozygous recessive at both. A dominant allele at any of the two loci originates a light purple color, while homozygous dominance produces dark purple color that superimposes to yellow, which is always present in the palisade

cells [5–7]. A more extended color classification has been advanced [6] which disregard green and brown seeds and broadly considers two levels of yellow and three of purple. However, with very few exceptions the most part of seeds of red clover fall in only two classes of color. In the case of the cultivar Altaswede used in this study these classes are the Classes 2 and 3 considered by Bortnem and Boe [6], which broadly equate with the terms yellow and light purple we adopted.

Seeds heteromorphism is known to affect a variety of physiological and ecological processes [9] throughout plants development and growth, namely during germination and early seedling growth [10,11]. These are undoubtedly the most vulnerable phases of plants' life cycle [12], in no small measure because once started seed germination is a critical and irreversible process committing the embryo to either death or to growth [13].

Recent research has shown that, in red clover seed, color also correlates with differential responses to water uptake, final germination and seedling emergence in soil, sensitivity of germination to salinity levels at or above 180 mM but not at or below 120 mM, with yellow seeds having the best performance, brown seeds the worst, and purple seeds intermediate or equal to yellow seeds in what concerns germination [7]. More recently, bright seeds from four different cultivars of red clover were shown to have higher germination rates and greater early seedling growth than dark seeds [14].

Thus, color seems to play a consistent role in the response of red clover seeds throughout imbibition, germination, and early seedling growth, which might provide guidelines for experimentally guided programs of breeding.

However some terminological questions exist in these studies, which raise some doubts regarding the consistency of a higher performance by the yellow morph, because 'bright' and 'dark' are unclear terms to describe red clover seeds. Bright seeds in [14] seem to equate with yellow seeds in [7], but it is unclear whether the dark seeds of the former refer to purple or brown seeds. As remarked above, brown is a color that results from seed ageing or from unfavorable conditions during production or storage, and if dark includes brown seeds, then the poor performance that was observed is justified per se.

Therefore, to evaluate whether or not the yellow morph performs consistently better than other morphs throughout the early stages of red clover life cycle, namely through imbibition, germination, and early seedling growth, we set out to investigate the effects of distilled water, NaCl, and complete nutrient solution. The latter was included as a surrogate for the soil solution, on seed germination, and early seedling growth of light purple and yellow morphs of red clover, separately investigating the three phases of germination characterized by [15], namely the sigmoidal phase of water imbibition, hereafter referred to as Phase I, the plateau phase of water imbibition hereafter referred to as Phase II, and the protrusion phase that visually allows for the recognition of seed germination, in the latter separating total germination from the path leading to it, namely lag of germination, time for final germination, duration of germination, and symmetry of the distribution of germination through time.

## 2. Materials and Methods

### 2.1. Plant Material

Seeds of *Trifolium pratense* L. cv. 'Altaswede' (Fabaceae) were provided by a specialized seed producer (Fertiprado Lda., Portugal), stored in closed paper bags under room conditions before being used in the experiments reported here, which were done around three years after seeds harvest.

The seeds were screened under a stereomicroscope Leica GZ4 and sorted according to the light purple or yellow color of their seed coat. Caution was exerted when sorting yellow morph, so that only pale and spotless seeds without strips or hues were retained. Color codes attribution was done by consensus. The RGB decimal color code of light purple morph was mainly 139, 20, 25 (HEX code #8B1419), and of yellow morph was mainly 255, 225, and 28 (HEX code #FFE11C).

Seeds biometry, including methods that were used for its determination, can be found in Appendix A below.

*2.2. Imbibition Experiment*

The experiment followed a dose-response design, with time of imbibition being the dose and weight increases at each time being the response, with a $2 \times 3$ arrangement of morph (light purple or yellow) and imbibing treatments (distilled water, NaCl solution, and complete nutrient solution).

The NaCl solution was prepared with distilled water and NaCl at 50 mM·L$^{-1}$ concentration. The complete nutrient solution was prepared from stock solutions, as described elsewhere [16], and was composed by 195.9 mg L$^{-1}$ N (as NO$_3$); 25.6 mg L$^{-1}$ N (as NH$_4$); 234.0 mg L$^{-1}$ K; 212.7 mg L$^{-1}$ Ca; 94.2 mg L$^{-1}$ Cl; 48.8 mg L$^{-1}$ Mg; 28.3 mg L$^{-1}$ P; 69.0 mg L$^{-1}$ S; 8.0 mg L$^{-1}$ Fe (chelated with EDTA); 1007.1 µg L$^{-1}$ B; 999.4 µg L$^{-1}$ Mn; 106.6 µg L$^{-1}$ Mo; 230.2 µg L$^{-1}$ Zn; 59.6 µg L$^{-1}$ Cu; 155.0 mg L$^{-1}$ Ni. Electrical conductivity and pH of solutions were determined with a conductivity meter Micro CM 2101 (Crison Instruments, SA, Barcelona, Spain) and a pH & Ion-Meter GLP 22 (Crison Instruments, SA, Barcelona, Spain), respectively. For each combination of morph, treatment, and time of imbibition, two glass Petri plates (diameter 3.5 cm) were fitted with a sponge 5 mm thick and wetted until saturation with distilled water or the appropriate solution. The whole was covered with Whatman No. 1 filter paper and 25 seeds were sown in a lattice pattern maximizing the distance among neighboring seeds. Throughout the experiment, the filter paper was kept permanently wet. Before being sown, each batch of 25 randomly selected seeds was individually weighed to the nearest mg with a scale Mettler Toledo PB153. The imbibing seeds were kept at dark under room conditions. Two additional batches of 25 randomly selected seeds per morph were individually weighed, as described (fresh weight), dried for 72 h at 80 °C in a fan-ventilated oven Cassel, allowed to cool under dry conditions, and dry weight was determined to the nearest mg with a scale Mettler Toledo PB153.

Air temperature of the room during the imbibition experiment was monitored at 15 min intervals with a data logger MicroLite I Temperature (Fourier Systems Inc., Mokena, IL, USA). Seeds of the two batches of each combination of morph and treatment were removed from Petri plates at two hour intervals, superficially dried with filter paper without pressing, and weighed to the nearest mg with a scale Mettler Toledo PB153. Weighing of seed batches in the imbibition experiment was done up to 18 h of imbibition, because, at 20 h, seed germination had already occurred in all Petri plates, which, together with high germination rates found in the germination experiment, strongly suggests that hardseededness if present, was not a meaningful factor. For unforeseen reasons, seeds weighing was not done at 12 h after the beginning of the experiment. The seeds were considered to be germinated when the embryo, usually the root, protruded the seed coat.

*2.3. Germination Experiment*

The designs of the imbibition experiment and of the germination experiment were essentially the same, with the time of imbibition being the dose and seed germination at each time being the response, with a $2 \times 3$ arrangement of morph (light purple or yellow) and imbibing treatments (distilled water, NaCl solution, and complete nutrient solution).

The differences were: in the germination experiment seeds were not weighed; four glass Petri plates instead of two were prepared for each combination of morph and treatment; seeds were daily examined for germination during 12 days; and, germinated seeds were removed.

*2.4. Seedling Growth Experiment*

The experiment followed a completely randomized design with a $2 \times 3$ arrangement of morph (light purple or yellow) and treatment (distilled water, NaCl solution, and complete nutrient solution).

The first seeds that completed germination in the seed germination experiment in distilled water, in NaCl solution, and in nutrient solution were transferred at 24 h, 30 h, or 30.5 h after sowing to newly prepared Petri plates that were fitted with Whatman No. 1 filter paper, wetted with distilled water, NaCl solution, or nutrient solution, kept under the same conditions as before, seedlings were allowed to grow for two days, and root and hypocotyl lengths were measured to the next mm.

## 2.5. Data Analyses

For each batch of 25 seeds, dryness (Ds) in percentage and imbibition (IMB) were determined as:

$$D_S = 100[(W_F - W_D)/W_D],\tag{1}$$

$$IMB = (W_a - W_b)/W_b,\tag{2}$$

where $W_F$ and $W_D$ are the fresh and dry weights, respectively, and $W_a$ and $W_b$ are the weights after and before imbibition, respectively. Replicates were defined by their rank of IMB-values [17], and the relation between IMB-values and time of imbibition was investigated using the four-term sigmoidal Weibull function [18] that was fitted by least squares nonlinear regression without replication using the Marquardt method [19].

The four-term Weibull function can be expressed as:

$$IMBT = M \{1 - \exp\{ - [(T - l)/k]^c\}\},\tag{3}$$

where IMBT is the cumulative imbibition at time T; $M$ is an asymptote for the cumulative imbibition and represents the maximum IMB-value for each batch of seeds; $l$ (lag of imbibition) is a location parameter that estimates the latest time at which imbibition is strictly zero, which in practice represents the time that is necessary for at least one seed start imbibing; $k$ (rate of imbibition) is a scale parameter estimating the rate of imbibition completion over time with $l + k$ estimating the time that is necessary for the completion of 63% of cumulative imbibition; and, $c$ is a dimensionless shape parameter estimating the symmetry of the distribution of imbibition over time, with $3.25 \leq c \leq 3.61$ showing symmetry and representing a good approximation to the normal distribution, $c < 3.25$ positive asymmetry, and $c > 3.61$ negative asymmetry [20–22]. The fitted equations were only accepted after a consistency check of parameter estimates and imbibition predictions against the original data [23]. The time necessary to reach $M$ ($T_M$) was derived from fitted equations, which allowed for the determination of the duration of imbibition ($D$) as $T_M$ minus $l$. All of the fitted and derived parameters of imbibing seeds treated with NaCl solution or with nutrient solution were separately compared for each morph with those that were obtained with distilled water, taken here as the control, by exact or approximate two-tailed Student's t tests after checking for homoscedasticity using the two-tailed F distribution. The same was done between the morphs for each treatment.

Contrary to the imbibition experiment, in which the maximum cumulative imbibition had to be estimated using Equation (3), in the germination experiment the maximum cumulative germination, hereafter referred to as final germination ($G_F$), was actually observed, which allowed, by making $M = 1$, the fitting of three-term Weibull equations by expressing cumulative germination at each time T in proportion of $G_F$. All other parameters of the Weibull equation were estimated or derived, as described above, with their meaning being the same with the necessary replacement of imbibition by germination.

Individual comparisons between treatments and distilled water for each morph and between the morphs were done as described for the imbibition experiment.

Root, hypocotyl, and total (root plus hypocotyl) lengths were divided by the time that the seedlings were allowed to grow after being transferred to new Petri plates.

For each morph, the root, hypocotyl, and total growth in NaCl solution or in nutrient solution were compared with growth in distilled water, taken here as control, by exact or approximate two-tailed Student's t tests after checking for homoscedasticity using the two-tailed F distribution.

The same was done for each treatment (distilled water, NaCl solution, and nutrient solution) between morphs.

Combinations of probabilities of independent Student's t tests was done using the inverse Chi-square method [24,25].

A comparison-wise error rate of $p = 0.05$ was used throughout and the data are presented as mean ± SE. Nonlinear regression analyses were done using Statgraphics Plus ver. 3.3 (Manugistics, Rockville, MD, USA); all other statistical analyses were executed using Microsoft Excel® 2010.

## 3. Results

Electrical conductivity and pH was 6.921 mS·cm$^{-1}$ and 5.56 in NaCl solution, and 2.346 mS·cm$^{-1}$ and 5.50 in nutrient solution. The air temperature ranged from 17.0 °C to 19.4 °C, with a mean ± SE of 18.1 ± 0.01 °C.

The dryness of seeds in the light purple morph was 13.6 ± 2.3%, being slightly higher but not significantly different ($p = 0.723$) than the yellow morph, which was 12.2 ± 2.8%.

The four-term Weibull equation could always be fitted to imbibition data and the coefficient of determination $R^2$ of fitted equations ranged from 0.919 to 0.994 with a mean of 0.964 ± 0.006. Almost without exception, treatments by the NaCl solution or by nutrient solution had no significant effects on the imbibition of seeds in comparison with distilled water either in light purple or in yellow morphs. The only exception was the duration of Phase I of the yellow morph treated with nutrient solution, which was significantly different and longer than in distilled water (Table 1C).

The three-term Weibull equation could also be always fitted to germination data and the $R^2$ of fitted equations ranged from 0.684 to 0.982 with a mean of 0.906 ± 0.013. Significant differences between treatments were found in the time that is necessary for the yellow morph to start and to finish germination in nutrient solution, which were longer than in distilled water (Table 1G,H) and in the final germination of light purple morph that was treated with nutrient solution, which was smaller than in distilled water (Table 1J).

Conversely, the growth of seedlings of light purple and yellow morphs was significantly affected by NaCl solution and by nutrient solution in comparison to distilled water (Table 1L,M) with root growth being reduced and hypocotyl growth increasing by both treatments, regardless of the morph.

Total growth, expressed by root plus hypocotyl growth, mirrored the significant effects that were observed in the response of root growth, except in the case of seedlings from the light purple morph that was treated with nutrient solution where no significant differences were found in relation to distilled water (Table 1N).

As for comparisons between morphs, no significant differences were found in any of the parameters of imbibition (Table 1A−F). In germination, significant differences were found in the time that is necessary for final germination and in the duration of germination of seeds that were treated with NaCl solution, with the light purple morph taking longer (Table 1H,I); in the final germination of seeds treated with nutrient solution with a lower rate of germination in the light purple morph (Table 1J); in the shape of germination of seeds in distilled water with a lower value in the light purple morph (Table 1K). In seedling growth, significant differences were always associated to treatments with nutrient solution with growth rates that were always higher in seedlings from the light purple morph (Table 1L−N).

Numbers that are inside parentheses are the significance levels of exact or approximate Student's t tests for differences between treatments and distilled water; numbers inside square parentheses (P) are the significance levels of exact or approximate Student's t tests for differences between light purple and yellow morphs for the same treatment

The results of a combination of probabilities of independent Student's t tests between morphs are summarized in Table 2. Significant differences between morphs were found in the overall experiment and in seeds that were treated by nutrient solution, but not in seeds treated by distilled water or NaCl. In relation to the three experiments, significant differences were only found in the germination and in the growth experiments.

**Table 1.** Means ± SE of seed imbibition (Phases I and II), seed germination, and early seedling growth of light purple and yellow morphs of seeds of *Trifolium pratense* cv. 'Altaswede' treated with distilled water, sodium chloride (salt solution), and complete nutrient solution.

| (A) Lag of Imbibition (h) | | | (B) Time for Maximum Imbibition (h) | | | (C) Duration of Phase I (h) | | |
|---|---|---|---|---|---|---|---|---|
| Treatment | Purple morph | P | Yellow morph | Purple morph | P | Yellow morph | Purple morph | P | Yellow morph |
| Control | 1.0 ± 0.04 | [0.496] | 2.0 ± 0.96 | 16.8 ± 1.7 | [0.740] | 16.0 ± 1.0 | 15.7 ± 1.8 | [0.502] | 14.0 ± 0.1 |
| Salt solution | 1.0 ± 0.03 (0.840) | [0.748] | 1.0 ± 0.04 (0.487) | 15.5 ± 0.5 (0.560) | [0.539] | 17.2 ± 2.2 (0.674) | 14.5 ± 0.5 (0.502) | [0.542] | 16.2 ± 2.2 (0.507) |
| Nutrient solution | 1.0 ± 0.03 (0.196) | [0.476] | 1.0 ± 0.01 (0.474) | 18.8 ± 0.7 (0.390) | [0.877] | 18.6 ± 0.6 (0.163) | 17.9 ± 0.8 (0.382) | [0.861] | 17.7 ± 0.6 (0.027) |

| (D) Maximum Imbibition (mg·mg$^{-1}$) | | | (E) Shape of Phase I | | | (F) Duration of Phase II (h) | | |
|---|---|---|---|---|---|---|---|---|
| Treatment | Purple morph | P | Yellow morph | Purple morph | P | Yellow morph | Purple morph | P | Yellow morph |
| Control | 1.2 ± 0.07 | [0.878] | 1.2 ± 0.02 | 2.3 ± 0.04 | [0.063] | 2.5 ± 0.01 | 3.2 ± 1.7 | [0.740] | 4.0 ± 1.0 |
| Salt solution | 1.1 ± 0.02 (0.626) | [0.673] | 1.1 ± 0.00 (0.073) | 2.4 ± 0.03 (0.355) | [0.221] | 2.5 ± 0.05 (0.646) | 4.5 ± 0.5 (0.560) | [0.539] | 2.8 ± 2.2 (0.674) |
| Nutrient solution | 1.1 ± 0.05 (0.898) | [0.273] | 1.0 ± 0.05 (0.131) | 2.4 ± 0.11 (0.485) | [0.718] | 2.5 ± 0.05 (0.901) | 1.2 ± 0.7 (0.390) | [0.877] | 1.4 ± 0.6 (0.163) |

| (G) Lag of Germination (days) | | | (H) Time for Final Germination (days) | | | (I) Duration of Germination (days) | | |
|---|---|---|---|---|---|---|---|---|
| Treatment | Purple morph | P | Yellow morph | Purple morph | P | Yellow morph | Purple morph | P | Yellow morph |
| Control | 0.7 ± 0.01 | [0.056] | 0.6 ± 0.03 | 4.3 ± 1.01 | [0.085] | 1.8 ± 0.14 | 3.6 ± 1.01 | [0.093] | 1.2 ± 0.11 |
| Salt solution | 0.5 ± 0.00 (0.538) | [0.163] | 0.4 ± 0.03 (0.426) | 3.2 ± 0.69 (0.642) | [0.036] | 1.6 ± 0.49 (0.144) | 2.7 ± 0.69 (0.643) | [0.038] | 1.2 ± 0.49 (0.399) |
| Nutrient solution | 1.0 ± 0.09 (0.060) | [0.949] | 1.0 ± 0.09 (0.009) | 4.6 ± 0.60 (0.814) | [0.169] | 3.4 ± 0.50 (0.019) | 3.6 ± 0.54 (0.991) | [0.146] | 2.4 ± 0.49 (0.055) |

| (J) Final Germination (%) | | | (K) Shape of Germination | | | | | |
|---|---|---|---|---|---|---|---|---|
| Treatment | Purple morph | P | Yellow morph | Purple morph | P | Yellow morph | | | |
| Control | 92.0 ± 1.6 | [0.494] | 95.0 ± 3.8 | 1.5 ± 0.39 | [0.030] | 2.8 ± 0.23 | | | |
| Salt solution | 92.0 ± 3.7 (1) | [0.718] | 94.0 ± 3.8 (0.859) | 1.0 ± 0.17 (0.967) | [0.079] | 1.8 ± 0.38 (0.399) | | | |
| Nutrient solution | 75.0 ± 3.0 (0.003) | [0.017] | 90.0 ± 3.5 (0.367) | 1.7 ± 0.17 (0.779) | [0.285] | 2.0 ± 0.25 (0.055) | | | |

| (L) Root Growth (mm·day$^{-1}$) | | | (M) Hypocotyl Growth (mm·day$^{-1}$) | | | (N) Root+hypocotyl growth (mm·day$^{-1}$) | | |
|---|---|---|---|---|---|---|---|---|
| Treatment | Purple morph | P | Yellow morph | Purple morph | P | Yellow morph | Purple morph | P | Yellow morph |
| Control | 5.4 ± 0.4 | [0.929] | 5.3 ± 0.4 | 1.6 ± 0.1 | [0.660] | 1.6 ± 0.1 | 7.0 ± 0.4 | [0.855] | 6.9 ± 0.4 |
| Salt solution | 1.5 ± 0.1 ($5.6 \times 10^{-11}$) | [0.175] | 1.7 ± 0.1 ($1.5 \times 10^{-8}$) | 3.7 ± 0.3 ($1.4 \times 10^{-8}$) | [0.128] | 3.2 ± 0.2 ($2.9 \times 10^{-8}$) | 5.2 ± 0.3 ($2.5 \times 10^{-4}$) | [0.363] | 4.9 ± 0.2 ($1.1 \times 10^{-4}$) |
| Nutrient solution | 2.1 ± 0.1 ($2.8 \times 10^{-9}$) | [0.005] | 1.6 ± 0.1 ($9.2 \times 10^{-9}$) | 4.3 ± 0.2 ($5.6 \times 10^{-12}$) | [0.004] | 3.3 ± 0.2 ($1.8 \times 10^{-7}$) | 6.4 ± 0.2 (0.185) | [$1.0 \times 10^{-4}$] | 4.9 ± 0.3 ($1.1 \times 10^{-4}$) |

**Table 2.** Values of the inverse Chi-square method ($X^2$), degrees of freedom and significance levels (P) for all comparisons, treatments, experiments, and treatments within experiments.

| Combinations of Probabilities of Comparisons between Morphs | $X^2$ | Degrees of Freedom | P |
|---|---|---|---|
| All comparisons | 139.2 | 84 | $1.4 \times 10^{-4}$ |
| Distilled water | 34.9 | 28 | 0.172 |
| Salt solution | 40.3 | 28 | 0.062 |
| Nutrient solution | 64.0 | 28 | $1.2 \times 10^{-4}$ |
| Imbibition experiment | 23.4 | 36 | 0.947 |
| Distilled water | 9.8 | 12 | 0.634 |
| Salt solution | 8.1 | 12 | 0.778 |
| Nutrient solution | 5.6 | 12 | 0.936 |
| Germination experiment | 64.6 | 30 | $2.5 \times 10^{-4}$ |
| Distilled water | 23.9 | 10 | 0.008 |
| Salt solution | 22.6 | 10 | 0.012 |
| Nutrient solution | 18.1 | 10 | 0.053 |
| Growth experiment | 51.2 | 18 | $1.2 \times 10^{-4}$ |
| Distilled water | 1.3 | 6 | 0.972 |
| Salt solution | 9.6 | 6 | 0.142 |
| Nutrient solution | 40.3 | 6 | $4.0 \times 10^{-7}$ |

Detailing more, no significant differences between the morphs were found in the imbibition experiment when distilled water, NaCl solution, and nutrient solution were considered separately. Conversely, significant differences between morphs were found in the germination experiment, but only in distilled water and in NaCl solution. The opposite was found in the growth experiment, with no significant differences in distilled water and NaCl solution, whereas significant differences between morphs were found with nutrient solution.

## 4. Discussion

Seeds of light purple and yellow morphs have similar and low dryness-values $D_S$, both below 14%, and thus qualify as dry seeds, which were defined as those with $D_S$ values lower than 15–20% [15].

Additionally, seeds of both morphs start imbibing very rapidly (1–2 h after being put in contact with water) and thereafter completed Phase I in less than 18 h, much less than in a number of other species belonging to Fabaceae [26–28] or to other families [29], where values ranging from a little more than one day to several days have been reported. Conversely, smaller durations of Phase I, lasting only 5 to 7 h, have been reported in Fabaceae [30] and in other families [31–33].

The values of the shape parameter of Phase I imply that the distribution over time of imbibition in distilled water or in the solutions investigated is positively asymmetric, with a pronounced tail to the right. Thus, in all likelihood, factors that govern the imbibition of red clover seeds are few, act multiplicatively [34], and are insensitive, either to differences between treatments and distilled water or between morphs.

On the contrary, Phase II was extremely short, lasting at most less than 5 h, regardless of treatment or morph. In addition, the duration of Phase II inversely varied with the duration of Phase I, with the former never lasting more than 30% of the latter, which is an exceptional result because Phase II in non-dormant seeds, like those investigated here, usually last much longer than Phase I [32].

In short, and with the single exception of nutrient solution in yellow morph, no effects of treatments or of seed coat color were found in the time for imbibition's triggering and in its subsequent unfolding throughout Phases I and II.

Because Phase I is largely a result of the high matric potential of seeds, regardless of seeds dormancy or viability [15], the absence of effects implies that the hydration and ability of cell walls,

starch, and protein bodies to bind water is essentially the same in the two morphs and is insensitive to differences in pH, osmotic potential, and composition of the solutions tested.

In Phase II, the matric potential of seeds loses its relevance to water uptake, while the water potential of seeds mostly results from the balance between osmotic potential and pressure potential [15]. Thus, not only do the two morphs complete, in a relatively short time, the metabolic events that are typical of Phase II that precede embryo protrusion, namely mitochondria repair and synthesis, as well as protein synthesis from existing and new mRNAs [35], but they also continue to appear insensitive to differences in pH, osmotic potential, and composition of solutions.

Contrary to Phase I and Phase II, only non-dormant seeds enter Phase III, which are usually equated with seed germination, a phase in which seeds swelling is visually superseded by the protrusion of seed coats by embryos usually starting with the root.

Final germination of red clover seeds was very high in distilled water and in NaCl solution, regardless of the morph, meaning that the very high rate of water uptake in the preceding phases did not result in noticeable imbibitional damage [15]. However, a significant reduction in final germination was found in seeds that were treated with nutrient solution, but only in the light purple morph, which in all likelihood should be attributed to the ionic composition of the solution.

Nevertheless, it has been shown that the effects of osmotic potential and pH on germination should be jointly considered [36], and therefore the interaction of effects between osmotic potential and pH cannot be completely put aside to explain the differential response of final germination of the light purple morph to NaCl and nutrient solutions in comparison to distilled water.

Significant differences in relation to distilled water were also found in the lag of germination and time that is necessary for final germination, but again only in seeds that were treated with nutrient solution. Opposite to what happened with final germination, it was the yellow morph that was affected.

Despite the effects on lag of germination that were observed in the yellow morph, the seeds of red clover start germinating in less than 24 h, which, according to [37], qualifies them as very fast germinating seeds (VFG), a feature that was not affected by treatments. The seeds of red clover essentially only partly fit the characteristics that are attributed to VFG [37,38]. In fact, they are not usually found in high stress habitats and they hardly qualify as very small, but they imbibe very rapidly and, being clovers, the endosperm is almost completely absent and the embryo occupies almost all seed [39].

There was an increase in the frequency of effects by treatments, which in germination occurred in 60% of parameters against 17% in Phases I and II. The yellow morph was clearly more sensitive to treatments than the light purple morph. In addition, and contrary to what happened in Phases I and II, significant differences between the morphs were found in the germination parameters in distilled water and NaCl solution, with the yellow morph requiring less or much less time for germination to start and to complete than the light purple morph, while simultaneously attaining higher values of final germination. Such better performance of the yellow morph, which was not observed in Phases I and II, has also been found, with few exceptions [40], in other studies involving a variety of cultivars of red clover [7,14,41], suggesting that seed coat color can be used as a broad correlate of seed germination vigor in red clover.

Finally, the values of the shape of germination imply that the distribution over time of water or solutions uptake is positively asymmetric, with a pronounced tail to the right. Thus, in all likelihood, whichever factors affected germination, they were few in number, acted multiplicatively [34], and were insensitive either to differences between solutions and distilled water or between morphs. However, the insensitivity of controlling effects between morphs was only found when the seeds were treated with NaCl or nutrient solutions, which appear to eliminate the differences between morphs found distilled water alone is involved.

The growth rate of seedlings was almost always strongly reduced by NaCl or by nutrient solution; with root growth being drastically reduced, while hypocotyl growth was stimulated in relation to distilled water. When reductions in relation to distilled water were observed, the effects were equally

intense in NaCl solution and nutrient solution or more intense in the former, which had a lower osmotic potential. Conversely, when increases in relation to distilled water were observed, the effects were equally intense in NaCl solution and nutrient solution or more intense in the latter, with the higher osmotic potential suggesting that, more than the single or joint effects of pH and osmotic pressure of solutions, the operative factors to early seedling growth responses might reside in the chemical composition of solutions.

In seedling growth, there was again an increase in the frequency of effects by treatments in relation to the preceding stage, and significant differences occurred almost in all parameters and comparisons. Conversely, differences between the morphs were less generalized than in germination, with the response of the two morphs only being different in nutrient solution and associated with a higher performance of the light purple morph instead of the yellow, as observed in germination. At first glance, the higher performance of light purple morph that is reported here contradicts the results from [14] with other cultivars of red clover. However, as noted above it is not clear whether dark seeds used by [14] were purple or brown. Therefore, if they were brown and not purple, then the higher performance of yellow seeds might essentially result from yellow seeds being younger than brown seeds [4].

## 5. Conclusions

The sensitivity of red clover seeds clearly increased with the stage of development in what seems to be a cumulative process from Phases I and II to seedling growth. In the former, no differences were found between morphs or between treatments; in the intermediate germination stage a noticeable increase in the frequency of effects by treatments was observed together with the appearance of differences between morphs in distilled water and in NaCl solution; finally, in seedling growth, the treatments were always effective, but differences between the morphs in distilled water and in NaCl solution were replaced by differences between morphs only in seeds that were treated by nutrient solution. In short, the superior performance of the yellow morph in relation to the light purple morph found in germination, which appears to be a trait stable across cultivars in red clover seeds, disappears as plants develop, and turns into a superior performance of the light purple morph in seedling growth.

**Author Contributions:** Conceptualization, A.S.D.; methodology, A.S.D.; validation, A.S.C. and A.S.D.; formal analysis, L.S.D.; investigation, A.S.C. and A.S.D.; resources, A.S.D.; writing—original draft preparation, L.S.D.; writing—review and editing, A.S.C., L.S.D. and A.S.D.; supervision, A.S.D.

**Funding:** This research received no external funding.

**Acknowledgments:** The authors thank Fertiprado Lda. for red clover seeds; the laboratories of Soil Chemistry, and of Plant Physiology, ICAAM, University of Évora for the facilities made available to us; Maria Gertrudes Grenho and Alexandra Cost, both affiliated to Department of Biology, University of Évora, for technical help in seeds biometry determinations, and for the participation in the RGB characterization of seeds respectively.

**Conflicts of Interest:** The authors declare no conflict of interest.

## Appendix A

Data on biometry of light purple and yellow morphs can be found in Table A1. One hundred seeds of each morph (light purple and yellow) were randomly selected and individual seed mass was determined to the nearest 0.1 mg with a scale Precise XR 2055M-DR, while length, width and thickness were individually determined using a digital caliper Comecta S.A. or a stereomicroscope Leica GZ4 equipped with an eyepiece micrometer Leitz Periplan $10 \times 18$ M under $10 \times 3$ total magnification.

Seed volume was determined from length, width and thickness measurements [42]. Seed density was derived from seed mass (g) and volume ($cm^3$). Seed sphericity was determined with data scaled to length as unity by dividing length, width and thickness by length [43]. For perspective, perfect sphericity equals zero while in the highly elongated caryopses of *Digitaria sanguinalis* ranges from 0.115 to 0.140 [42]. Light purple seeds are larger, heavier and denser than yellow seeds, traits identified

since long in the seeds of red clover [40,41] despite that differences of seed weight might depend on the location within plants of the flowers bearing the seeds [40].

**Table A1.** Ranges, means ± SE and medians of mass, volume, density, and sphericity of light purple and yellow morphs of seeds of *Trifolium pratense* cv. 'Altaswede'.

| Characteristic | Parameters | Purple Morph | Yellow Morph |
|---|---|---|---|
| Seed mass (mg) | Range | 0.6–18.0 | 1.3–2.8 |
| | Mean ± SE | 2.1 ± 0.17 | 1.7 ± 0.03 |
| | Median | 1.8 | 1.7 |
| Seed volume (mm$^3$) | Range | 3.000–10.243 | 3.040–9.956 |
| | Linear dimensions (mm) | $3.22 \times 2.17 \times 0.82$–$4.27 \times 3.85 \times 1.19$ | $2.87 \times 2.38 \times 0.85$–$4.90 \times 2.94 \times 1.32$ |
| | Mean ± SE | 6.793 ± 0.131 | 6.405 ± 0.138 |
| | Median | 6.887 | 6.244 |
| Seed density | Range | 0.186–2.369 | 0.181–0.428 |
| | Mean ± SE | 0.302 ± 0.021 | 0.279 ± 0.004 |
| | Median | 0.281 | 0.275 |
| Seed sphericity | Range | 0.081–0.114 | 0.069–0.116 |
| | Mean ± SE | 0.098 ± 0.001 | 0.096 ± 0.001 |
| | Median | 0.098 | 0.097 |

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
