# Peer review of "Imbibition, Germination, and Early Seedling Growth Responses of Light Purple and Yellow Seeds of Red Clover to Distilled Water, Sodium Chloride, and Nutrient Solution"

_sci, doi:10.3390/sci1020051_

Round 1

Reviewer 1 Report

The paper “imbibition, germination, and early seedling growth responses of light purple and yellow seeds of red clover to distilled water, sodium chloride, and nutrient solution” provides further evidence that red clover seed coat color is related to seed germination.  The paper is well written and I have only a few comments I feel the authors should address.

The paper classifies red clover seed as being either yellow, light purple, or dark purple.  This classification is consistent with Nijdam (1932).  However, Bortnem and Boe (2000) propose a five-class color classification scheme.  Based on images presented in Bortnem and Boe (2003) I tend to agree that more color classification categories are needed than three.  The authors should address this critique somehow in the paper.  There use of yellow and “light purple” seed for the study is still valid as everyone agrees on what constitutes yellow seed.  It would be interesting to see if the authors repeated this study using “dark purple” seed.  “Altaswede” according to Bortnem and Boe is known to have significantly lighter seed than “Arlington” a North American variety (see images in paper). 

Please provide the age of the seed used.   How many days or months after seed harvest was seed used in this experiment?

Red clover seed often displays hardseededness; this trait prevents seed from imbibing water at all.  Hardseed in red clover is controlled by the seed production environment and genetics.  What efforts did the authors of this study make to determine the amount of hardseed in the seed lot obtained?  Was hardseed observed?

Author Response

The paper “imbibition, germination, and early seedling growth responses of light purple and yellow seeds of red clover to distilled water, sodium chloride, and nutrient solution” provides further evidence that red clover seed coat color is related to seed germination. The paper is well written and I have only a few comments I feel the authors should address. The paper classifies red clover seed as being either yellow, light purple, or dark purple. This classification is consistent with Nijdam (1932). However, Bortnem and Boe (2000) propose a five-class color classification scheme. Based on images presented in Bortnem and Boe (2003) I tend to agree that more color classification categories are needed than three. The authors should address this critique somehow in the paper. There use of yellow and “light purple” seed for the study is still valid as everyone agrees on what constitutes yellow seed. It would be interesting to see if the authors repeated this study using “dark purple” seed. “Altaswede” according to Bortnem and Boe is known to have significantly lighter seed than “Arlington” a North American variety (see images in paper). Please provide the age of the seed used. How many days or months after seed harvest was seed used in this experiment? Red clover seed often displays hardseededness; this trait prevents seed from imbibing water at all. Hardseed in red clover is controlled by the seed production environment and genetics. What efforts did the authors of this study make to determine the amount of hardseed in the seed lot obtained? Was hardseed observed? Dear Dr Riday On behalf of all authors let me thank for the considerations, comments and suggestions made to our submission. A detailed point by point reply goes below and we changed the manuscript accordingly. Please notice that in addition to the changes suggested by you the revised version presents other changes suggested by the other reviewer. Hopefully you will not disagree with them or at least not very strongly. Sincerely Luís Review Report Form English language and style ( ) Extensive editing of English language and style required ( ) Moderate English changes required (x) English language and style are fine/minor spell check required ( ) I don't feel qualified to judge about the English language and style Yes Can be improved Must be improved Not applicable Does the introduction provide sufficient background and include all relevant references? ( ) (x) ( ) ( ) Is the research design appropriate? (x) ( ) ( ) ( ) Are the methods adequately described? ( ) (x) ( ) ( ) Are the results clearly presented? (x) ( ) ( ) ( ) Are the conclusions supported by the results? (x) ( ) ( ) ( ) Comments and Suggestions for Authors The paper “imbibition, germination, and early seedling growth responses of light purple and yellow seeds of red clover to distilled water, sodium chloride, and nutrient solution” provides further evidence that red clover seed coat color is related to seed germination. The paper is well written and I have only a few comments I feel the authors should address. The paper classifies red clover seed as being either yellow, light purple, or dark purple. This classification is consistent with Nijdam (1932). However, Bortnem and Boe (2000) propose a five-class color classification scheme. Based on images presented in Bortnem and Boe (2003) I tend to agree that more color classification categories are needed than three. The authors should address this critique somehow in the paper. There use of yellow and “light purple” seed for the study is still valid as everyone agrees on what constitutes yellow seed. It would be interesting to see if the authors repeated this study using “dark purple” seed. “Altaswede” according to Bortnem and Boe is known to have significantly lighter seed than “Arlington” a North American variety (see images in paper). RE: Bortnem and Boe proposed a five color classification (disregarding greens and browns) but as far as we understood their paper the large majority of seeds belong to at most three classes (2, 3 and 4) which represent between 82 and 97% of seeds. The only exception in their data is Arlington M258-WA AL8. However differences exist among cultivars/lots. In a number of them the top colors are 3 and 4. In others the top colors are 2 and 3. The latter occurs in Altaswede, which seem to correspond broadly to what we considered yellow and light purple, making the two colors we tested the best representative colors of the cultivar. We added a period on this matter to the third paragraph of the introduction. Please let us know if this period corresponds to what you had in mind in your remark. We agree that it would be interesting to have used “dark purple” seeds but at the time it was not possible to secure the work and logistic required. Please provide the age of the seed used. How many days or months after seed harvest was seed used in this experiment? RE: Added as requested. Red clover seed often displays hardseededness; this trait prevents seed from imbibing water at all. Hardseed in red clover is controlled by the seed production environment and genetics. What efforts did the authors of this study make to determine the amount of hardseed in the seed lot obtained? Was hardseed observed? RE: We did not specifically studied the amount of hard seeds but we had to stop the imbibition experiment at 18 hours because at 20 hours all seeds had germinated. Also we found very high rates of germination (mean values of 92% in purple, 95% in yellow). Altogether we feel that this supports that hardseededness, if present at all, was a negligible factor. We added a brief note on this in the end of imbibition experiment description in Materials and Methods. We think it suffices but please let us know if you think a more elaborate reasoning is necessary. Submission Date 24 January 2019 Date of this review 11 Jun 2019 00:08:12

Reviewer 2 Report

The paper “Imbibition, Germination, and Early Seedling Growth Responses of Light Purple and Yellow Seeds of Red Clover to Distilled Water, Sodium Chloride, and Nutrient Solution” is well-written, the goal is transparent and methods used are quite simple, but definitively labour-intensive.

Some minor comments are stated below.

I) Please, add to “Materials and Methods”:

a) the content of nutrient solution used in the experiment;

b) a few methodical words of explanation about the shape of phase 1 and the shape of germination.

II) Please, summarise the results in the conclusions based on the research question from “Introduction” (the beginning of the last paragraph): “… to evaluate whether or not the yellow morph performs consistently better than all other morphs throughout the early stages of red clover life cycle, namely through imbibition, germination, and early seedling growth …”

III) Some editorial mistakes need correction.

a) Please, check the second paragraph in “2.4.Seedling Growth Experiment”: “at 24 h, 30 h, or 30h30 after …” – is “30h30” correct?  

b) Please, check comas in references, they should appear after the name of the author – nr 7: it should be “Mavi, K.”, nr 35: it should be “Dias, L.S.”.

Author Response

The paper “Imbibition, Germination, and Early Seedling Growth Responses of Light Purple and Yellow Seeds of Red Clover to Distilled Water, Sodium Chloride, and Nutrient Solution” is well-written, the goal is transparent and methods used are quite simple, but definitively labour-intensive. Some minor comments are stated below. I) Please, add to “Materials and Methods”: a) the content of nutrient solution used in the experiment; b) a few methodical words of explanation about the shape of phase 1 and the shape of germination. II) Please, summarise the results in the conclusions based on the research question from “Introduction” (the beginning of the last paragraph): “… to evaluate whether or not the yellow morph performs consistently better than all other morphs throughout the early stages of red clover life cycle, namely through imbibition, germination, and early seedling growth …” III) Some editorial mistakes need correction. a) Please, check the second paragraph in “2.4.Seedling Growth Experiment”: “at 24 h, 30 h, or 30h30 after …” – is “30h30” correct? b) Please, check comas in references, they should appear after the name of the author – nr 7: it should be “Mavi, K.”, nr 35: it should be “Dias, L.S.”. Dear Dr Hanaka On behalf of all authors let me thank for the considerations, comments and suggestions made to our submission. A detailed point by point reply goes below and we changed the manuscript accordingly. Please notice that in addition to the changes suggested by you the revised version presents other changes suggested by the other reviewer. Hopefully you will not disagree with them or at least not very strongly. Sincerely Luís Review Report Form English language and style ( ) Extensive editing of English language and style required ( ) Moderate English changes required (x) English language and style are fine/minor spell check required ( ) I don't feel qualified to judge about the English language and style Yes Can be improved Must be improved Not applicable Does the introduction provide sufficient background and include all relevant references? (x) ( ) ( ) ( ) Is the research design appropriate? (x) ( ) ( ) ( ) Are the methods adequately described? ( ) (x) ( ) ( ) Are the results clearly presented? (x) ( ) ( ) ( ) Are the conclusions supported by the results? ( ) (x) ( ) ( ) Comments and Suggestions for Authors The paper “Imbibition, Germination, and Early Seedling Growth Responses of Light Purple and Yellow Seeds of Red Clover to Distilled Water, Sodium Chloride, and Nutrient Solution” is well-written, the goal is transparent and methods used are quite simple, but definitively labour-intensive. Some minor comments are stated below. I) Please, add to “Materials and Methods”: a) the content of nutrient solution used in the experiment; RE: Added as requested. b) a few methodical words of explanation about the shape of phase 1 and the shape of germination. RE: The shape of phase I and the shape of germination refer and are described by the c parameter of the Weibull function and, as stated in Materials and Methods, in germination has the same meaning it had when the Weibull equation was described in relation to imbibition in the preceding paragraph. That is to say that it refers to the symmetry/asymmetry of imbibition or germination through time. Because it was explained for imbibition few lines above we think that it is not necessary to repeat the explanation, which would be essentially the same just replacing imbibition by germination. Hopefully we made it clearer by rephrasing the 3rd paragraph of page 4. II) Please, summarise the results in the conclusions based on the research question from “Introduction” (the beginning of the last paragraph): “… to evaluate whether or not the yellow morph performs consistently better than all other morphs throughout the early stages of red clover life cycle, namely through imbibition, germination, and early seedling growth …” RE: Rephrased. III) Some editorial mistakes need correction. a) Please, check the second paragraph in “2.4.See.dling Growth Experiment”: “at 24 h, 30 h, or 30h30 after …” – is “30h30” correct? RE: Yes, it is correct. Seedlings used in this experiment were those that appeared in the germination experiment. Sowing of seeds in the germination experiment was not completely simultaneous and picking and transferring them neither, so there was some delay in that, which did not occurred in the first seedlings that were transferred after 24 hours. b) Please, check comas in references, they should appear after the name of the author – nr 7: it should be “Mavi, K.”, nr 35: it should be “Dias, L.S.”. RE: Done. Also throughout the paper. Submission Date 24 January 2019

Round 2

Reviewer 1 Report

The revisions have improved the manuscript.

Author Response

See below. On behalf of the authors thank you for all the time and attention given to our submission. Sincerely Luís

Reviewer 2 Report

Dear Authors,

Corrections are well done. But please, explain clearly "30h30". Does it mean "30 h 30 min"?

With regards.

Author Response

See below. We changed "30h30" to "30.5 h", as is usually done despite mixing two numerical systems (10-based and 60-based). On behalf of the authors thank you for all the time and attention given to our submission. Sincerely Luís